# Soil nitrogen concentration mediates the relationship between leguminous trees and neighbor diversity in tropical forests

Han Xu (ID) et al.#

Legumes provide an essential service to ecosystems by capturing nitrogen from the atmosphere and delivering it to the soil, where it may then be available to other plants. However, this facilitation by legumes has not been widely studied in global tropical forests. Demographic data from 11 large forest plots (16–60 ha) ranging from 5.25° S to 29.25° N latitude show that within forests, leguminous trees have a larger effect on neighbor diversity than non-legumes. Where soil nitrogen is high, most legume species have higher neighbor diversity than non-legumes. Where soil nitrogen is low, most legumes have lower neighbor diversity than non-legumes. No facilitation effect on neighbor basal area was observed in either high or low soil N conditions. The legume–soil nitrogen positive feedback that promotes tree diversity has both theoretical implications for understanding species coexistence in diverse forests, and practical implications for the utilization of legumes in forest restoration.

---

#A list of authors and their affiliations appears at the end of the paper.

The beneficial effect of nitrogen (N)-fixing legumes on neighbors is commonly exploited in agricultural systems by using intercropping practices[1,2]. However, outside of agricultural systems, the influence of N-fixing legumes on the surrounding plant community is poorly understood. Studies of legumes have reported both positive effects[3–5] or negative effects[6,7] on biomass and growth. N-fixing legumes can also potentially influence local species composition and diversity, but the direction and the mechanisms of such an influence remain unclear[8,9].

One possible explanation for neighbor facilitation by legumes is that biological N fixation (BNF) favors mycorrhizal colonization and soil microbial activity[10]. These microbial symbionts may help mineralize organic phosphorous (P), mitigating the chronic phosphorous limitation of many tropical ecosystems[1,11] and releasing soil nutrients from mineral to more soluble-forms, especially rhizosphere soil N[12–14]. This activity might increase complementary resource use (i.e., alleviate niche overlap) between legumes and neighbors[9,15], thereby promoting neighbor biomass, growth, and diversity.

On the other hand, the competitive advantage of legumes over neighbors and the fertilization effect of N-rich litter may promote competitive exclusion, a strong destabilizing mechanism that reduces diversity. The competitive advantage of legumes varies across different forests and it is mediated by their divergent BNF strategies (namely facultative fixation, obligate fixation, and non-fixation), which depends on soil nutrient supply and nutrient demands at different growth stages[14,16–19]. For example, conventional wisdom suggests that late-successional tropical forest soils, which are rich in N but limited in P, should restrict the competitive advantage of N-fixing plants[20]. This trend is supported by the high rates of nodulation and fixation of legumes observed in N-poor and recently disturbed forests[21,22]. In tropical forests, legume abundance measured by absolute and relative basal area is highest during the early stages of forest succession and is often higher in dry, as compared to wet, secondary forests[23]. However, a high number of N-fixing legume species are also found in undisturbed tropical forests[24] and there is little evidence of decline in N-fixer abundance in forest chronosequence studies[8,25,26]. Legume richness and abundance also increases from subtropical to tropical areas[14,27,28], but across the tropics their distribution varies widely, with higher abundance in the Neotropics and Africa than in South East Asia[29].

The inconsistent findings on the competitive advantages and abundance of legumes in different forest regions are possibly caused by species-specific differences in the amount and timing of N-fixation[30], growth demand for leaf N, photosynthetic capacity, and water-use efficiency[16]. Moreover, not all legume species have the potential for BNF[17], and many legume species are facultative N-fixers[31], meaning they only fix N under certain environmental conditions (e.g., N-poor soils). A key element to consider, therefore, is that legumes play a dual role as soil nutrient demanders and providers. This feature could potentially explain the contrasting effects on neighborhood tree diversity and biomass, depending on how many abiotic (e.g., soil N, P concentration) and biotic factors (e.g., fungal network connection) interact to influence tree neighborhood interactions. Thus, the effect of legumes on neighbor diversity and biomass in highly diverse forests still lacks a clear consensus, both in terms of direction and magnitude.

In order to establish if general patterns exist for the interactions between leguminous trees and their neighbors, we used data from 11 large tropical and subtropical forest dynamics plots (16–60 ha, Fig. 1) where all individuals are mapped, identified to species, and repeatedly measured the diameter at the breast height every 5 years. We ask two questions. (1) Do legumes facilitate neighbor diversity or increase neighbor biomass in tropical forests? (2) Do the interactions of leguminous trees with their neighbors vary in relation to soil N concentration, soil P concentration, local temperature and precipitation? If legumes support greater neighbor tree diversity when compared to non-legumes, and this positive effect on tree diversity is mediated by soil N concentration, this suggests that legumes actively promote a positive belowground feedback that acts as a stabilizing mechanism for neighbor tree diversity. This positive feedback contrasts with certain negative belowground feedbacks, such as soil pathogens and other soil microbiome negative density-dependent forces, that are thought to be stabilizing mechanisms that constrain neighbor diversity of plant communities[32,33].

An implication of the importance of legumes to tropical forest communities is that the variation in soil N may reflect legume abundance, because of the high tissue N concentration in leguminous trees. Legumes improve soil N by returning N-rich organic litter to the soil, which may help maintain soil N pools in tropical forests[14]. To understand the specific factors that enable legumes to facilitate or inhibit local diversity in a particular tropical environment, we also investigated how local interactions of these leguminous trees relate to their capacity to fix di-nitrogen gas ($N_2$). For each of the 272 legume species present in the 11 large forest plots, we assessed their potential to fix $N_2$ based on

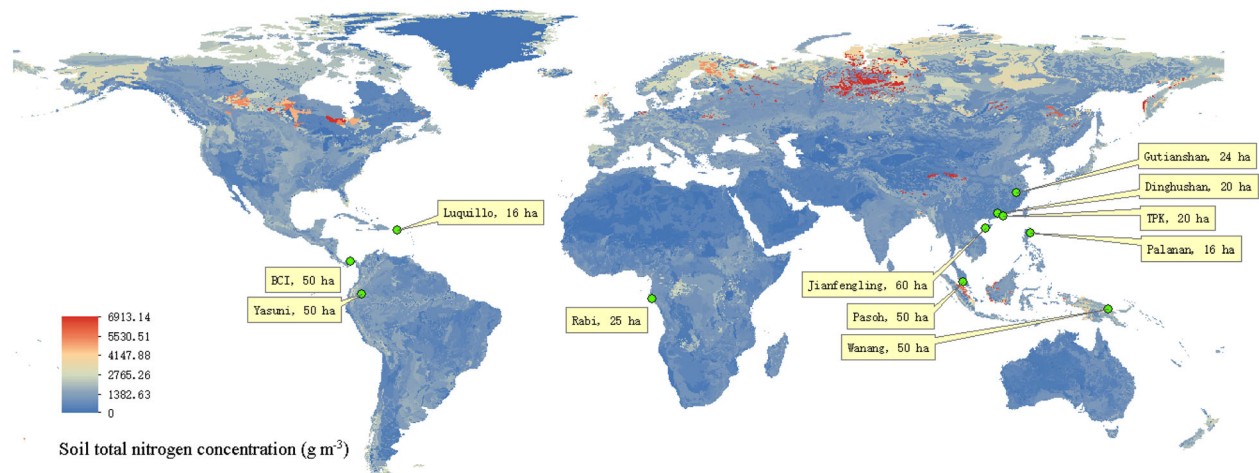

**Fig. 1 Global map showing the location of the 11 Forest-GEO plots.** This map is overlaid with the IGBP-DIS 1-m depth soil total N concentration (5 × 5 arc-minute resolution)[42].

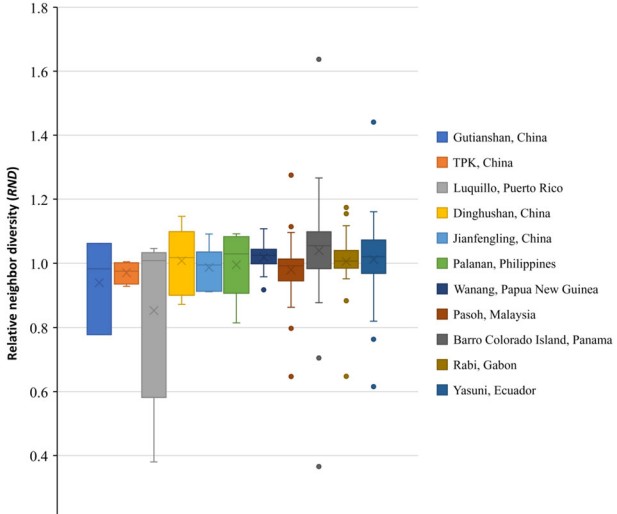

**Fig. 2 Boxplot of legume relative neighbor density at a 2 m radius from the focal legume for 11 ForestGEO plots.** Relative neighbor density (RND) is the ratio of the diversity of legume neighborhoods to the diversity of non-legume neighborhoods. It is displayed in order of number of legume species in each forest plot.

nodulation reports and expert knowledge[34]. As expected, we found that soil nitrogen concentration mediates the relationship between leguminous trees and neighbor diversity.

## Results

**Neighbor diversity and basal area of all legume species**. Results showed that the effect of leguminous trees on relative neighbor diversity (RND) compared to non-legumes varied across the forests, which can be positive (RND > 1) or negative (RND < 1) for different legumes in each plot at 2 or 4 m radius (Fig. 2, Supplementary Fig. 3). The relative neighbor density at 2 m has more variation than that at 4 m radius. The multiple regression analysis showed that, of the factors we tested soil total N concentration was the only factor that significantly correlated with the proportion of legume species with positive RND in each plot (Supplementary Table 1), and no environmental variables significantly correlated with the proportion of legume species with positive relative neighbor basal area (RNBA) in each plot (Supplementary Table 2). We also did not find any significant relationships between the number of either legume or non-legume species, and soil total N concentration ($p > 0.05$, Fig. 3a, b), although both variables significantly decreased with latitude ($R^2 = 0.8883$ and 0.7164, both with $p < 0.001$, Fig. 3c, d).

**Soil nitrogen mediates the neighbor diversity of all legumes**. Across all 11 forests, the proportion of legume species with positive RND showed a linear response to increases in soil total N concentration ($R^2 = 0.8364$, $p < 0.001$, Fig. 4a) and this was not a random effect (Supplementary Fig. 1), but showed no significant distributional patterns with respect to latitude ($p > 0.05$, Fig. 4c), precipitation, temperature, or soil total P concentration (all with $p > 0.05$). This trend was strong at a small neighborhood radius from focal individuals (2 m) and quickly vanished at larger radii ≥4 m (Supplementary Fig. 2). Therefore, in forests with low soil N concentration, focal legumes showed lower immediate neighbor diversity than did non-legume taxa, but in forests with higher soil N concentration immediate neighbor diversity was higher than for non-legume trees.

Across the 11 forests, the proportion of legume species with positive RNBA did not show significant responses to soil total N

concentration ($p > 0.05$, Fig. 4b), latitude ($p > 0.05$, Fig. 4d), nor the other three environmental factors (precipitation, temperature, or soil total P concentration, all with $p > 0.05$).

**Soil nitrogen mediates the neighbor diversity of N-fixing legumes**. When only potentially N-fixing legumes were considered, the proportion of N-fixing legume species with positive RND still showed a relatively strong association with variation in soil total N concentration ($p = 0.097$, Fig. 5a). The decay of the association is mainly caused by one plot (Rabi) which has a much lower proportion of N-fixers, and where almost 91% of the legume species are non-N-fixers, with only four N-fixing species. When data from the Rabi plot were removed, the proportion of legume species with positive RND still showed a significant and strong response to soil total N concentration gradients ($p = 0.0017$, Fig. 5b).

## Discussion

The analyses presented here took advantage of 11 tropical forest plots between 5.25°S and 29.25°N latitude with a large variation in legume richness from 3 to 104. Our results support that variation in soil N concentration mediates the relationship between leguminous trees and neighbor diversity, but not basal area. In forests with low soil total N concentration, legumes do not promote the growth or diversity of immediate neighbors. In forests with high soil total N concentration (above 1122 g m$^{-3}$), a high proportion of (>53%) number of legume species exhibit a positive feedback on local diversity, but they do not promote biomass accumulation. Previous studies have also reported that N-fixer abundance has no effect on biomass recovery during forest succession and can even inhibit growth in regenerating rainforests in Panama and Costa Rica[6,7].

In soils with high N concentration, the facilitating effect of legumes on neighbors may be caused not only via bacterial fixation of N from the atmosphere[14], but also through N cycling via ectomycorrhizal fungi. These microorganisms use the organic rather than inorganic soil nutrients and compensate for the lack of soil N supply, even in the absence of fixation, as expected in soil with high N levels[13,35]. These properties increase the competitive ability of legumes and promote resource partitioning (niche divergence) along soil N gradients between legumes and non-legumes[9]. In contrast, in low soil N environments the presence of N-fixing legumes likely intensifies competition, thereby reducing neighbor diversity. This suggests that, under low soil N conditions, legumes may not share biologically fixed N, consistent with the low-N tolerance life style, which implies slow N recycling rates[9,31]. In one N addition experiments for three legumes species, we also found that the rhizosphere soil nitrogen can be significantly lower than that of non-rhizosphere. The differences between rhizosphere and non-rhizosphere even varied with the gradients of N addition concentration. In general, these results reveal that legumes may be 'altruistic' or 'self-serving' and do not facilitate others with a purpose, which should be taken into account when legumes are used for ecosystem restoration because of the consideration that the "direct transfer" of N from fixers to the non-fixers, as in plantation intercropping practices and regeneration forests[1,36].

This study illustrates the important role of legumes in maintaining high levels of soil N in many of the world's tropical forests[14], although we did not directly measure the N fixation ability or and N demand of legume species for each plot. Not all legumes are N-fixers, and the proportion of N-fixers in the legume community of the 11 plots in this study varies from 11% to 100%, based on the nodulation fixer list[34,37]. For the other 10 plots excluding Rabi plot, the proportion of N-fixing legumes of

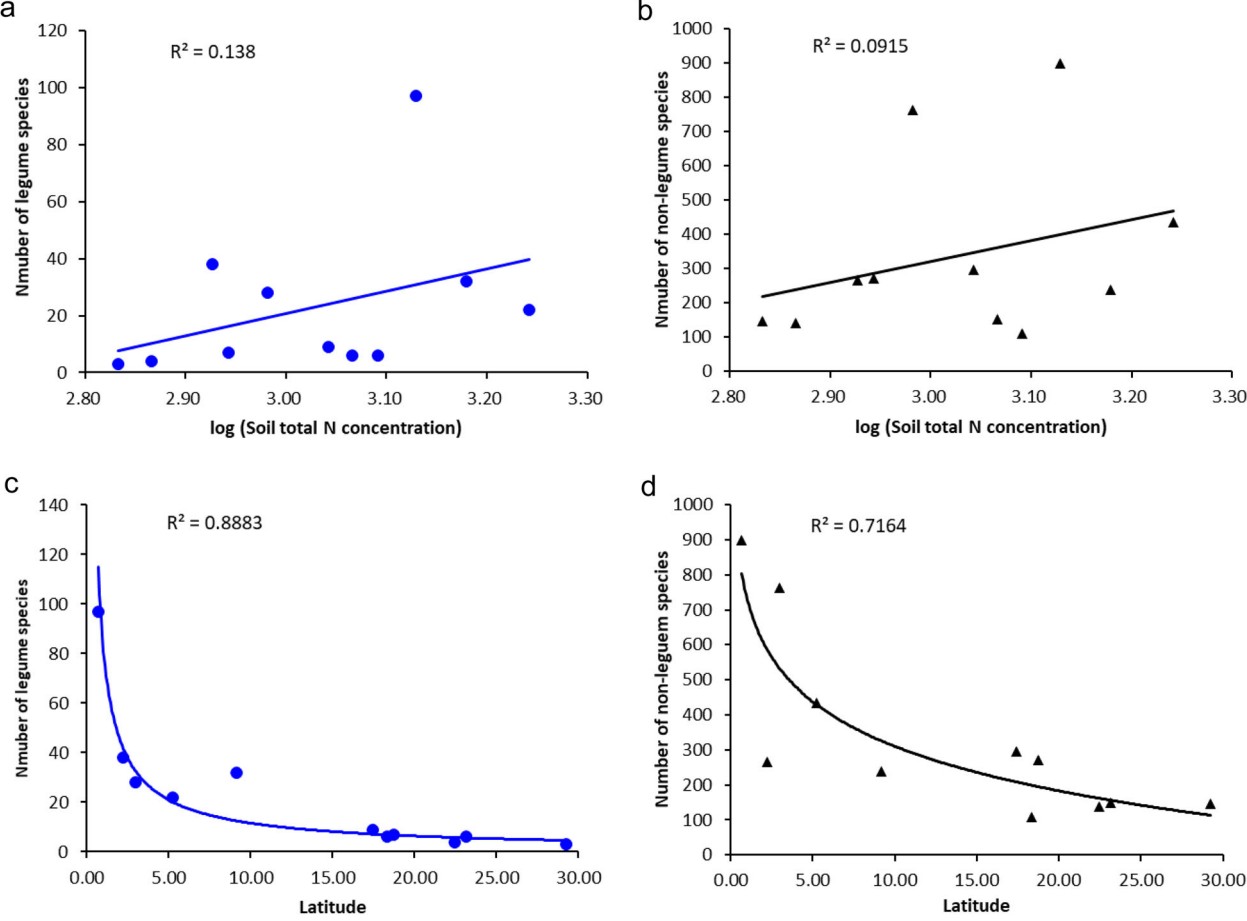

**Fig. 3 Abundances of legume and non-legume species for 11 ForestGEO plots across the gradients in soil total N concentration (g m$^{-3}$) and latitude.** **a** Number of legume species and **b** number of non-legume species in relation to soil total N concentration (g m$^{-3}$). **c** Number of legume species and **d** number of non-legume species in relation to latitude.

total legume species varied from 32% to 100%. The much lower proportion of N-fixers in the Rabi plot infers that the N-fixing process may not be the first factor driving the legume species distribution and coexistence in this plot, which can alternatively be temperature, precipitation, or even topographical factors.

The maintenance of tree diversity and productivity reflects interactions among all tree species, not only legumes. Across the 11 forest plots (Table 1) studied here and other forests globally[14], legumes occupy between 0.8% and 74.4% of plot basal area. Therefore, the effects of legumes on the forest ecosystem processes vary among sites and across the globe, being amplified by the latitudinal gradient in legume tree abundance (Fig. 3c, d). More detailed analyses within these plots are also needed, for example, using locally measured data on the spatial variation in soil nutrients such as N and P to evaluate variation of N-fixer neighborhood effects.

Most legumes hold higher N concentration in their tissues (leaves, stems, and seeds) than non-legumes[16], although they do not always have the highest tissue N concentrations in the community[9]. A large amount of organic litter N will return back to the soil and improve soil N gradually during and after the lifetime of an N-fixing tree, especially in the case of species with high foliar N concentration, which preferentially grow in high soil N habitats[9]. This process may asymmetrically increase the soil N levels in the habitats where there is already a relatively high soil N concentration. It is inferred that, when legumes are alive, most of them play

important roles as soil N demanders, rather than providers that satisfy their growth demands in high soil N habitats.

In sum, we first found that leguminous trees do not necessarily facilitate neighbor plant diversity in the tropical forests, and leguminous trees and neighbors' association is mediated by the local soil N concentration across a wide latitudinal gradient. To further strictly establish neighbor beneficial relations for individual trees, we need to assess if the growth or survival rates of legume neighbors are higher than those neighbors around non-legumes using tree growth monitoring data in each plot. This analysis would provide a direct assessment of facilitation effects of legumes on immediate neighbors. We can also ask how different legumes are influenced by their neighbors because of their varied nutrient requirements, rhizosphere, and microbiome effects, which requires the collection of legume-specific traits and intensify of the soil sampling within plot. Additional future research could investigate how rates of N fixation and enrichment of soil N around legumes interact with tree community demographics, and assesses the rates of asymbiotic N fixation by free-living soil bacteria, which is comparable to the symbiotic N fixation by legumes and may be another important source of soil N[10,38,39]. This might help to explain the lack of correlation between soil N content and number of legume species richness and abundance, and why the competition in low soil N levels only reduces neighbor tree diversity but not basal area. Understanding different sources of soil N especially mycorrhizal associations, will provide further insights into the spatial interactions of legume and non-legume trees in tropical forests.

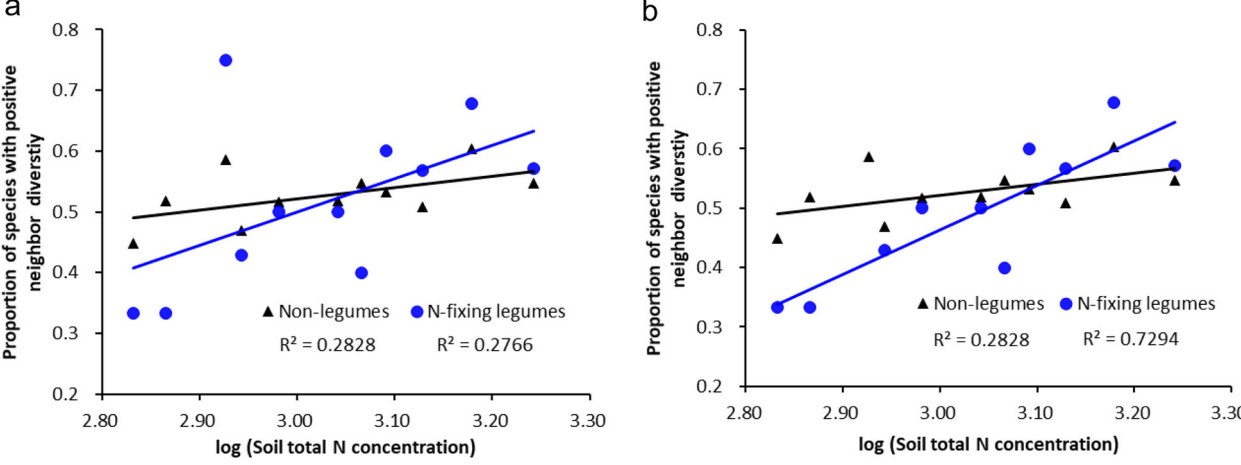

**Fig. 4 Proportion of legume and non-legume species with positive neighbor diversity (RND) and positive neighbor basal area (RNBA) in relation to soil total N concentration (g m$^{-3}$) and latitude.** Data shown for 11 ForestGEO plots at a spatial radius of $r = 2$ m. Proportion of legume and non-legume species with positive **a**, RND and **b**, RNBA in relation to soil total N concentration (g m$^{-3}$), respectively; and with positive **c**, RND and **d**, RNBA in relation to latitude, respectively.

**Fig. 5 Proportion of N-fixing legume and non-legume species with positive neighbor diversity (RND) in relation to soil total N concentration (g m$^{-3}$) with different numbers of plots at a spatial radius of $r = 2$ m. a** 11 plots. **b** 10 plots.

## Methods

**Statistics and reproducibility**. We carried out tree censuses in 11 forest plots widely distributed from latitude 5.25°S to 29.25°N that included the principal tropical forest formations of the world (Table 1, Fig. 1). Only one census data per plot was used in this study. The area of the plots ranged from 16 to 60 ha, which is larger than typical forest inventory data and it is designed to include, for in each site, most of the local species and reflect the local variations in species composition and structure. All the plot censuses followed a standard enumeration and included all stems with diameter at the breast height (DBH diameter ≥1 cm at 1.3 m above the ground as required from the ForestGEO plot network, https://www.forestgeo.si.edu/). All stems were identified to species, measured and mapped to their relative spatial coordinates within each forest plot[40,41].

**Table 1 Basic information for 11 forest plots.**

| No. | Plot name and country | Latitude | Longitude | Plot area (ha) | Total number of all species | Total number of legume species | Total basal area of all species (m²) | Total basal area of legume species (m²) | Total individuals of all species | Total individuals of legume species | Soil total N concentration (g m⁻³) | Soil total P concentration (g m⁻²) | Annual temperature (°C) | Annual precipitation (mm) |
|---|---|---|---|---|---|---|---|---|---|---|---|---|---|---|
| 1 | Gutianshan, China | 29.25 | 118.117 | 24 | 156 | 3 | 886 | 7 | 140,700 | 718 | 680.2 | 207.0 | 15.9 | 1430 |
| 2 | TPK, China | 22.4263 | 114.181 | 20 | 167 | 5 | 586 | 5 | 117,203 | 601 | 734.0 | 312.0 | 23.0 | 2334 |
| 3 | Luquillo, Puerto Rico | 18.3262 | −65.816 | 16 | 129 | 6 | 578 | 44 | 46,360 | 2119 | 1234.4 | 272.4 | 25.6 | 2363 |
| 4 | Dinghushan, China | 23.1695 | 112.511 | 20 | 168 | 7 | 504 | 5 | 79,603 | 3399 | 1164.3 | 321.9 | 22.0 | 1870 |
| 5 | Jianfengling, China | 18.7308 | 108.905 | 60 | 283 | 7 | 3314 | 36 | 391,686 | 10,447 | 877.1 | 485.2 | 24.9 | 2102 |
| 6 | Palanan, Philippines | 17.4402 | 122.388 | 16 | 314 | 9 | 709 | 19 | 74,426 | 1226 | 1102.3 | 253.5 | 25.1 | 2724 |
| 7 | Wanang, Papua New Guinea | −5.25 | 145.267 | 50 | 556 | 25 | 1603 | 111 | 276,139 | 9665 | 1745.8 | 363.9 | 26.5 | 3366 |
| 8 | Pasoh, Malaysia | 2.982 | 102.313 | 50 | 818 | 28 | 1549 | 126 | 335,347 | 9155 | 958.9 | 312.0 | 26.3 | 1896 |
| 9 | Barro Colorado Island (BCI), Panama | 9.1543 | −79.8461 | 50 | 267 | 35 | 1502 | 163 | 210,814 | 16,463 | 1510.5 | 604.5 | 26.3 | 3025 |
| 10 | Rabi, Gabon | −2.2228 | 9.88004 | 25 | 303 | 43 | 740 | 250 | 175,661 | 30,017 | 845.3 | 604.5 | 26.1 | 1943 |
| 11 | Yasuni, Ecuador | −0.6859 | 76.397 | 50 | 966 | 104 | 820 | 122 | 145,823 | 18,259 | 1345.8 | 207.0 | 25.9 | 3270 |

Annual temperature and precipitation of each plot were obtained from the basic site information of the CTFS-ForestGEO network as described by Anderson-Teixeira et al. 2015 (climate data for 1980–2012 from CGIAR-CSI climate data)[41]. It is displayed in order of number of legume species in each plot.

Soil total N density (g m⁻³) data were obtained from the global gridded surfaces of selected soil characteristics (IGBP-DIS) database at a resolution of 5 × 5 arc-minutes and for the soil depth interval 0–100 cm from Oak Ridge National Laboratory Distributed Active Archive Center (https://daac.ornl.gov) for each plot[42].

Soil total P (g m⁻²) data were obtained from the global gridded soil phosphorus distribution maps at 0.5° resolution database for the soil depth interval 0–50 cm from Oak Ridge National Laboratory Distributed Active Archive Center (ORNL DAAC, https://daac.ornl.gov) for each plot[43].

Ideally, local comparable soil N and P data from the same depth should be used, but these were only available for three sites (BCI, Gutianshan, Jianfengling). For those sites, the local soil nutrients in the 0–10 cm layer and those obtained from the global dataset were positively correlated (Pearson's $r = 0.84$, $p < 0.001$ for soil total N; Pearson's $r = 0.63$, $p < 0.001$ for soil total P), indicating that global datasets is a good approximation to the soil N and P contents in these three plots (Supplementary Table 3). The soil nutrients from the global database also reflect the long-term soil evolution process and can better represent the landscape conditions. Therefore, only the soil N and P nutrient data from these two databases are used for this comparative analysis. Also, this study neglects the local heterogeneity in soil N and P contents and focuses on average values to reflect the large-scale patterns. Therefore, a unique value of soil N and P contents for each plot is obtained from the global ORNL DAAC database.

Annual temperature and precipitation of each plot were obtained from the climate information available for each site of the CTFS-ForestGEO network as described by Anderson-Teixeira et al.[41].

Canopy openness may be another abiotic factor that affects soil nutrients, species composition and competition among plants, especially within each forest. Unfortunately, a reliable measure of light availability and canopy openness for the whole forest plot is not available. These plots are located in tropical and subtropical mature forests, where canopy openness is low and gaps are ephemeral. Therefore, any variation in the forest light environment within forests is not likely adequately sampled, given the large size of the CTFS-ForestGEO plots (i.e., all plot > 20 ha in size).

**Species neighbor diversity calculation**. To evaluate whether the presence of a particular species has a significant effect on local diversity, we used a marked point pattern statistical analysis, based on the individual species–area relationship or ISAR[9,44]. ISAR is defined as the diversity (D), or basal area (the indicator of biomass, BA) for all neighbor species within distance r from an individual of a focal species i:

$$\text{ISAR}_i(r) = \frac{1}{N_i} \sum_{j \in i} S_{ij}(r) \quad (1)$$

where $S_{ij}(r)$ is the observed Shannon-equivalent diversity index (exponential Shannon) or basal area of neighbor species around the individual j of focal species i within distance r from the focal individual, and $N_i$ is the number of individuals of focal species i. The analysis is restricted to focal individuals and neighbors with DBH ≥1 cm. Species richness generally varies with latitude and biomass is a function of site-specific climatic and edaphic conditions. To test if ISAR was significantly higher or lower than ISAR of neighbor species, and make the statistics comparable among sites, we computed a RND and RNBA as the ratio:

$$\text{RND}_i(r) \text{ or } \text{RNBA}_i(r) = \frac{\text{ISAR}_i(r)}{\frac{1}{N_i} \sum_{j \in i} H_{ij}(r)} \quad (2)$$

where $H_{ij}$ is the average Shannon-equivalent diversity index or basal area of neighbor species computed around all matching non-focal species that grow within 60 m from a focal individual and in the same DBH class. The choice of 60 m is a compromise between the need to include enough non-focal individuals with different neighborhoods, and keeping the area relatively small to control for the large-scale habitat heterogeneity within each plot[9]. Within each plot, DBH is equally classified into 20 logarithmic size classes.

Because RND and RNBA are ratios, 1 was regarded as the cut-off point to define if they are positive (>1) or negative (between 0 and 1). $\text{RND}_i(r) > 1$ or $\text{RNBA}_i(r) > 1$ indicate that an individual of a focal species i has, on average, higher diversity or basal area within a radius of distance r than an individual of a non-focal species with similar DBH that grows in the same area, i.e. the focal species is an "attractor". Conversely, $\text{RND}_i(r) < 1$ or $\text{RNBA}_i(r) < 1$ indicates that the focal species is a "repeller". We evaluated the RND and RNBA to compare focal trees with DBH size matched non-focal trees.

RND and RNBA rapidly approach one, with increasing distance r from the focal individuals. At large distances the plant neighborhoods of each different species become indistinguishable. Therefore, in order to assess the immediate neighbor effects of focal legumes and non-legumes, with neighbor trees, we only report the results at the distance r = 2 and 4 m.

**Statistical analysis**. First, to reveal which of the environmental variables tested might best explain the relative neighborhood effects of legumes among forests, we used a multiple regression to model the relationship between the proportion of legume species with positive RND and RNBA (with $\text{RND}_i(r) > 1$, and $\text{RNBA}_i(r) > 1$) and the environmental factors of each plot, including soil total N concentration, soil total P concentration, latitude, annual temperature, and annual precipitation.

The proportion of legume species with positive neighbor richness ($RND_i(r) > 1$) and basal area ($RNBA_i(r) > 1$) were plotted against the significant environmental factors and fitted with linear models. We quantified how legume abundance, in term of species number, varied across latitude and along the soil total N gradients. We also calculated the proportions of legume and non-legume species with positive RND and RNBA for all focal legume and non-legume species in each plot.

**Reporting summary**. Further information on research design is available in the Nature Research Reporting Summary linked to this article.

## Data availability

All plots data can be acquired from the ForestGEO plot network (https://www.forestgeo.si.edu/). Soil N and P contents for each plot can be obtained from the global ORNL DAAC database. Data for Fig. 2–5 is provided as a Source Data file.

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

## Acknowledgements

This work was supported by NSFC (31670628, 31370441) and National Non-profit Institute Research Grant of CAF (CAFYBB2017ZE001). We thank all persons, institutions, and funds in the Supplementary Note 1 to support the establishment of ForestGEO dynamic plots across the whole world before we could use the data in this paper.

## Author contributions

H.X., M.D., and S.F. planned experiments, analyzed data and wrote the paper together with other co-authors. R.C., J.A.H., R.V., J.K.Z., J.T, A.A., and S.L.Y. revised the paper substantially. Y.L., B.C.H.H., G.A.F., G.D.W., M.U., J.L., K.C., D.K., P.B., H.R.M., S.J.D., X.M., and T.L.Y. also helped to collect the data and analyzed the data. Tree measurement data are contributed from all co-authors from 11 forest plots.

## Competing interests
The authors declare no competing interests.

## Additional information

Han Xu [1,23], Matteo Detto [2,3,23], Suqin Fang [4✉], Robin L. Chazdon [5], Yide Li[1], Billy C. H. Hau [6], Gunter A. Fischer[7], George D. Weiblen[8], J. Aaron Hogan [9,10], Jess K. Zimmerman[11], Maria Uriarte [12], Jill Thompson[13], Juyu Lian[14,15], Ke Cao[16], David Kenfack [3], Alfonso Alonso[17], Pulchérie Bissiengou[18], Hervé Roland Memiaghe[19], Renato Valencia[3,20], Sandra L. Yap[21], Stuart J. Davies[3], Xiangcheng Mi[16] & Tze Leong Yao [22]

[1]Research Institute of Tropical Forestry, Chinese Academy of Forestry, 510520 Guangzhou, China. [2]Department of Ecology and Evolutionary Biology, Princeton University, Princeton, NJ, USA. [3]Forest Global Earth Observatory (ForestGEO), Smithsonian Tropical Research Institute, 20013 Washington, DC, USA. [4]State Key Laboratory of Biocontrol, School of Life Sciences, Sun Yat-Sen University, 510275 Guangzhou, China. [5]Department of Ecology and Evolutionary Biology, University of Connecticut, Storrs, CT, USA. [6]School of Biological Sciences, The University of Hong Kong, Hongkong, China. [7]Flora Conservation Department, Kadoorie Farm and Botanic Garden, Hongkong, China. [8]Bell Museum, University of Minnesota, 140 Gortner Laboratory,1479 Gortner Avenue, Saint Paul, MN 55108, USA. [9]International Center for Tropical Botany, Florida International University, Miami, FL 33199, USA. [10]Ecosystem Science Division, Oak Ridge National Laboratory, Oak Ridge, TN 37831, USA. [11]Department of Environmental Science, University of Puerto Rico - Río Piedras, San Juan, PR 00931, USA. [12]Department of Ecology, Evolution and Environmental Biology, Columbia University, 1200 Amsterdam Avenue, New York, NY 10027, USA. [13]UK Centre for Ecology & Hydrology, Bush Estate, Penicuik, Midlothian EH26 0QB, UK. [14]Key Laboratory of Vegetation Restoration and Management of Degraded Ecosystems, South China Botanical Garden, Chinese Academy of Sciences, Tianhe District Guangzhou, China. [15]Guangdong Provincial Key Laboratory of Applied Botany, South China Botanical Garden, Chinese Academy of Sciences, Tianhe District, Guangzhou, Guangdong, China. [16]State Key Laboratory of Vegetation and Environmental Change, Institute of Botany, Chinese Academy of Sciences, 20 Nanxincun, 100093 Xiangshan Beijing, China. [17]Center for Conservation and Sustainability, Smithsonian Conservation Biology Institute, Washington, DC, USA. [18]Herbier National du Gabon, Libreville, Gabon. [19]Institut de Recherche en Écologie Tropicale, Libreville, Gabon. [20]Escuela de Ciencias Biológicas, Pontificia Universidad Católica del Ecuador, Apartado, 17-01-2184 Quito, Ecuador. [21]Institute of Arts and Sciences, Far Eastern University, Manila, Philippines. [22]Forest Research Institute Malaysia, 52109 Kepong, Selangor, Malaysia. [23]These authors contributed equally: Han Xu, Matteo Detto. ✉email: fangsuq5@mail.sysu.edu.cn

