## [Peer Review File · Communications Biology]

Reviewers' comments:

Reviewer #1 (Remarks to the Author):

This study examines the facilitation effect of leguminous tree species on both neighbor diversity and basal area relative to the non-legume ones. Tropical forests are used as a study system, with data from 11 large forest plots ranging from 5.25° S to 29.25° N latitude being used.

This paper presents an important issue that has been mostly studied in grasslands and agricultural systems.

There are some concerns about the data and depth of the analysis. In particular:

(a) I am concerned about the use of various scales of soil nutrient data (with a different resolution and depth of measurement) in relation to the size of the forest plots (16-60ha). Nitrogen density data ($\text{g}\cdot\text{m}^{-3}$) were obtained at a resolution of 5x5 arc-minutes and soil depth interval of 0-100cm, phosphorus concentration data ($\text{g}\cdot\text{m}^{-2}$) at 0.5-degree and soil depth interval of 0-50cm. Significant positive correlation between local soil nutrient data for three sites and those obtained from global dataset have been provided. To the best of my understanding, authors calculate correlations using soil nutrient concentrations obtained from different depths (local soil nutrients were measured at a depth of 0-10cm), different reference area and without inform us about the number of replicates used in this analysis.

(b) All graphs and statistics has been done with 11 numbers, without true replication. This does not guarantee a robust statistical analysis.

(c) Authors have no taken into account the light availability (canopy openness) that affect soil moisture and, along with soil N, functional composition and diversity of natural plant communities.

The neighbor effect of focal legume species with neighbor trees is reported at the distance $r = 2$ m and $r = 4$ m. Have we any idea whether belowground N-transfer from leguminous trees to non-leguminous trees is restricted to 4m? In that point, studies using ^{15}N labeled leguminous trees could be used to justify this choice.

Figure 2 could be also depicted at a 4 m radius, and some comparison with $r = 2$ m could be made.

Reviewer #2 (Remarks to the Author):

The manuscript by Xu, et al. uses tree census data from 11 large forest plots, widely distributed with respect to latitude, to statistically analyze how legume and non-legume tree

species differently affect their neighboring tree diversity and relative biomass, taking into consideration variables including soil nitrogen and phosphorous concentrations, latitude and annual temperatures and precipitation. The main questions that the study wished to answer are clearly stated on page 7, to paraphrase: 1. In tropical forests, do legumes increase neighbor diversity or biomass? 2. How are the legume-neighbor interactions influenced by soil N and P concentration and climatic factors? Hypotheses are offered as to how soil N could increase or constrain neighbor diversity of legumes. This focus on soil N is relevant because the analysis presented shows that the relative neighbor diversity of legumes correlated positively with soil N, while relative neighbor basal area (roughly equivalent to biomass) did not correlate with any of the variables. This supports the notion that high soil N influences legume tree neighbor diversity but not biomass and that in forests with low soil N, the legumes do not promote diversity or biomass of neighbors. This seems to fit in with cited literature showing that the abundance of nitrogen fixers does not effect biomass recovery in specific forest successions, and is probably inhibitory to rainforest regeneration in some locations. Depending on soil N conditions, legumes can facilitate or hamper diversity, which should be taken into account in planning reforestation efforts and plantation intercropping systems. The manuscript describes published information or reasonable suppositions that relate to the findings, such as how legume derived fixed nitrogen can affect the soil microbiota and how storage of N in legume leaves can determine their roles as N demanders or providers at different stages of their life cycle. At the end of the manuscript, some topics for future analysis and refinement of these results are presented, along with a restatement of the major finding, that "...leguminous trees do not necessarily facilitate neighbor diversity on tropical forests, and leguminous trees and neighbors' association if mediated by local total N concentration...."

I find the work convincing up to the point that the available data allow conclusions to be drawn: as I mentioned, additional data that would strengthen and extend these conclusions is mentioned in the text. I'm no expert on stats, but from what I read on multiple regression it appears to be a valid approach to analyzing this type of multi-variable data and drawing conclusions with a certain probability of being "true". I also think the paper will have a positive influence not only for the conclusions drawn but also in directing additional research. The manuscript is generally well written, brief but seemingly complete as regards the literature review and analysis and discussion of the results obtained. Perhaps in the future it would also be interesting to identify specific symbiotic and associative nitrogen fixers in soil (or obtained from nodules or the rhizosphere/root surfaces, as the case may be), as well as other microbiota, to see if this might help in explaining the non-correlation between soil N content and number of legume species and total legume species individuals in the different plots

Michael F. Dunn
Centro de Ciencias Genómicas-UNAM
Cuernavaca, Morelos, Mexico

Reviewer #3 (Remarks to the Author):

Summary:

Using the measurements done in plots located in 11 tropical forests and publicly available soil data, the authors tried to show that soil N will mediate the relationship between leguminous trees and neighbor diversity in tropical forests. The major conclusion is that "Where soil nitrogen is high, most legume species have higher neighbor diversity than non-legumes. Where soil nitrogen is low, most legumes have lower neighbor diversity than non-legumes." (lines 64-66). While it is a potentially interesting finding, the conclusions could have been oversimplified whereas a lot of variable factors have been ignored. Some major concerns are listed below.

Comments:

(1) One major concern is the grouping of all legume trees into one single factor. Some of the sampled legume trees are even not nitrogen-fixing (lines 185-186). For instance, when all legume trees are considered, the significance of the correlation between positive RND and soil N has a p-value of 0.0252 (Extended Data Table 1); however, if only nitrogen-fixing legume trees are considered, the p-value is 0.097 (line 189). Technically, 0.097 is insignificant if the cut off is set to the regular 0.05 level. Does it imply that the nitrogen-fixing ability of the legume trees is not an important factor in this research?

(2) Furthermore, the nitrogen-fixing capacity, growth rate, growth character (e.g. shading effects), nutrient requirements (competition effects), microbes at the rhizosphere could be different from one species to another. These factors are ignored in this analysis. How to control for these factors?

(3) Legumes occupy between 0.8% and 74.4% of plot basal area (line 197). Will the RND calculation be affected by this factor. For example, when comparing 0.8% occupancy versus 74.4% occupancy, will the estimation of the diversity of non-leguminous trees be affected?

(4) The comparison of RND among different plots could be problematic. The "non-legume neighborhoods" could be very different in different plots, since the non-legume tree species will be different in different latitudes. Will different non-legume trees in the "non-legume neighborhoods" exert differential effects on the diversity? How to control for this factor?

(5) The soil P and N data are based on the average data from an archive which used a very large grid compared to the current study. Local P and N data in the plots may have changed due to the species in the legume neighborhoods and non-legume neighborhoods. Local P and N data in the plots should be used.

(6) The authors reported that in 3 plots, "the local soil nutrients in the 0-10 cm layer and those obtained from the global dataset were positively correlated" (lines 243-244). How about the actual data (not just the correlation)? Are they similar and to what degree are the

error margins?

(7) The authors concluded that "variation in soil N concentration mediates the relationship between leguminous trees and neighbor diversity but not basal area." (lines 160-161). "In soils with high N concentration.....but also through N cycling via ectomycorrhizal fungi.....in low soil N levels the presence of N-fixing legumes likely intensifies competition, thereby reducing neighbor diversity." (lines 168-175).

First, different legume trees will have different associated ectomycorrhizal fungi, the N cycling abilities and the effects on neighbor diversity should be validated if not documented before. Second, why is the competition in low soil N levels only reduce neighbor diversity but not basal area?

(8) The authors should add sub-headings to the sections Introduction, Results, and Discussions.

Responses to comments

Reviewer #1 (Remarks to the Author):

This study examines the facilitation effect of leguminous tree species on both neighbor diversity and basal area relative to the non-legume ones. Tropical forests are used as a study system, with data from 11 large forest plots ranging from 5.25° S to 29.25° N latitude being used. This paper presents an important issue that has been mostly studied in grasslands and agricultural systems. There are some concerns about the data and depth of the analysis. In particular:

(a) I am concerned about the use of various scales of soil nutrient data (with a different resolution and depth of measurement) in relation to the size of the forest plots (16-60ha). Nitrogen density data (g.m⁻³) were obtained at a resolution of 5x5 arc-minutes and soil depth interval of 0-100cm, phosphorus concentration data (g.m⁻²) at 0.5-degree and soil depth interval of 0-50cm. Significant positive correlation between local soil nutrient data for three sites and those obtained from global dataset have been provided. To the best of my understanding, authors calculate correlations using soil nutrient concentrations obtained from different depths (local soil nutrients were measured at a depth of 0-10cm), different reference area and without inform us about the number of replicates used in this analysis.

R: In the methods, we have described the details of the soil sampling design and measurements of nutrients in the Jiangfengling forest plot. Previous studies in that plot found that the soil N contents at different depths, 0-10 cm, 10-30 cm and 30-60 cm, are significantly related with R² values of 0.46 (between 0-10 cm and 10-30 cm), 0.28 (between 10-30 cm and 30-60 cm), 0.49 (between 0-10 cm and 30-60 cm) and $p < 0.05$ (Shi LL, 2012, Master thesis of the Chinese Academy of Forestry). Ideally, if soil nutrient data could have been obtained from each of the 11 plots individually, it would be better. However, not all plots had measured the local soil nutrient variation. Yet, local soil nutrients in each plot can be quite variable. By using interpolated soil nutrients data from the global databases, we aim to provide a larger scale average conditions of nutrients levels that reflect the distribution of legumes across global gradients.

The data from the Oak Ridge National Laboratory Distributed Active Archive Center (ORNL DAAC, <https://daac.ornl.gov>) represent a well-maintained standard data source for global soil nutrient data. Although it does not contain the information about the sampling

intensities, there is extensive metadata and other information on the data available through the ORNL DAAC data portal. Only one soil nutrient value per plot is used in this study. Several sentences are added and revised to the methods part to illustrate it clearly (L. 254-264).

(b) All graphs and statistics has been done with 11 numbers, without true replication. This does not guarantee a robust statistical analysis.

R: One of the main goal of the Smithsonian Forest-GEO network (<https://forestgeo.si.edu/>) is to perform cross-site comparisons using forest data from large forest dyanics plots cesnsused using standardized methods. The statndardized methodology allows for global comparison of plots, wherin plots are treated as repilciates. We emphasize that the size of all Forest-GEO plots (20-60 ha) is much larger than typical plot size (which is usually about 1 ha), making each a representative sample of its respective forest community. Therefore, the Forest-GEO sapling design reduces uncertainties related to local variation in species composition by measuring hundreds of thousands of individual plants in each plot. We have added one sentence to the methods part to explain it clearly (L. 239-241).

With these considerations in mind, we agree that 11 independent points do not guarantee robust statistical analysis and for all the plots we now report p -values to provide the level of confidence on the strength of the correlations. For the significant relationship with nitrogen, the p -value was lower than 0.05 (i.e. confidence level >95%).

(c) Authors have no taken into account the light availability (canopy openness) that affect soil moisture and, along with soil N, functional composition and diversity of natural plant communities.

R: We agree that light availability is an important abiotic factor that affects species composition and competition among plants, especially within each forest. Unfortunately, a reliable measure of light availability and canopy openness is not available to well represent the light conditions of each Forest-GEO plot with such such large area. Furthermore, the plots are located in the tropical and subtropical mature forests, where canopy openness is low and gaps are ephemeral. Therefore, any variation in the forest light environment within forests is not likely adequately sampled, given the large size of the CTFS-ForestGEO plots (i.e., all plot > 20 ha in size). We are also not sure how much variability in legumes-neighbors interactions could be explained by canopy openness. We have added this consideration in the methods part (L268-274).

The neighbor effect of focal legume species with neighbor trees is reported at the distance $r = 2$ m and $r = 4$ m. Have we any idea whether belowground N-transfer from leguminous trees to non-leguminous trees is restricted to 4m? In that point, studies using ^{15}N labeled leguminous trees could be used to justify this choice.

R: In a previous study (Xu, Detto et al., 2019) we used nitrogen isotopes to explore the effect of legumes on local neighbors. That study motivated the global analysis presented in the current manuscript. We used N^{15} labeled methods to detect if the nitrogen fixation of different legumes was related to neighbor richness. Results showed that it was indeed the case and neighbors of legumes with high nitrogen fixing abilities showed more positive ^{15}N isotopes signature in the leaf material. Although from that study it was not possible to determine the exact range of the interactions, results showed that the effect of legumes on neighbors declined quickly with distance.

Figure 2 could be also depicted at a 4 m radius, and some comparison with $r = 2$ m could be made.

R: One more figure (Supplementary Fig. 3) is added at the 4m radius in the Supplementary files part. The comparison with $r=2$ m is also made (L. 139-141).

Reviewer #2 (Remarks to the Author):

The manuscript by Xu, et al. uses tree census data from 11 large forest plots, widely distributed with respect to latitude, to statistically analyze how legume and non-legume tree species differently affect their neighboring tree diversity and relative biomass, taking into consideration variables including soil nitrogen and phosphorous concentrations, latitude and annual temperatures and precipitation. The main questions that the study wished to answer are clearly stated on page 7, to paraphrase: 1. In tropical forests, do legumes increase neighbor diversity or biomass? 2. How are the legume-neighbor interactions influenced by soil N and P concentration and climatic factors? Hypotheses are offered as to how soil N could increase or constrain neighbor diversity of legumes. This focus on soil N is relevant because the analysis presented shows that the relative neighbor diversity of legumes correlated positively with soil N, while relative neighbor basal area (roughly equivalent to biomass) did not correlate with any of the variables. This supports the notion that high soil N influences legume tree neighbor diversity but not biomass and that in forests with low soil N, the legumes do not promote diversity or biomass of neighbors. This seems to fit in with cited literature showing that the abundance of nitrogen fixers does not effect biomass recovery in specific forest successions, and is probably inhibitory to rainforest regeneration in some locations. Depending on soil N conditions, legumes can facilitate or hamper diversity, which should be taken into account in planning reforestation efforts and plantation intercropping systems. The manuscript describes published information or reasonable suppositions that relate to the findings, such as how legume derived fixed nitrogen can affect the soil microbiota and how storage of N in legume leaves can determine their roles as N demanders or providers at different stages of their life cycle. At the end of the manuscript, some topics for future analysis and refinement of these results are presented, along with a restatement of the major finding, that “...leguminous trees do not necessarily facilitate neighbor diversity on tropical forests, and leguminous trees and neighbors’ association is mediated by local total N concentration....”

I find the work convincing up to the point that the available data allow conclusions to be drawn: as I mentioned, additional data that would strengthen and extend these conclusions is mentioned in the text. I’m no expert on stats, but from what I read on multiple regression it appears to be a valid approach to analyzing this type of multi-variable data and drawing conclusions with a certain probability of being “true”. I also think the paper will have a positive influence not only for the conclusions drawn but also in directing additional research. The manuscript is generally well written, brief but seemingly complete as regards the

literature review and analysis and discussion of the results obtained. Perhaps in the future it would also be interesting to identify specific symbiotic and associative nitrogen fixers in soil (or obtained from nodules or the rhizosphere/root surfaces, as the case may be), as well as other microbiota, to see if this might help in explaining the non-correlation between soil N content and number of legume species and total legume species individuals in the different plots.

Michael F. Dunn

Centro de Ciencias Genómicas-UNAM

Cuernavaca, Morelos, Mexico

R: We thank the reviewer for the accurate summary of key findings of this study and the positive feedback.

As suggested, we have added that future study should focus on identifying the specific symbiotic and associative nitrogen fixing microbial groups in soil which might help to explain the lack of correlation between soil N content and number of legume species or the legume abundance in the different plots. This suggestion has been added to the final paragraph of discussion (L. 228-231).

Reviewer #3 (Remarks to the Author):

Summary: Using the measurements done in plots located in 11 tropical forests and publicly available soil data, the authors tried to show that soil N will mediate the relationship between leguminous trees and neighbor diversity in tropical forests. The major conclusion is that “Where soil nitrogen is high, most legume species have higher neighbor diversity than non-legumes. Where soil nitrogen is low, most legumes have lower neighbor diversity than non-legumes.” (lines 64-66). While it is a potentially interesting finding, the conclusions could have been oversimplified whereas a lot of variable factors have been ignored. Some major concerns are listed below.

Comments:

(1) One major concern is the grouping of all legume trees into one single factor. Some of the sampled legume trees are even not nitrogen-fixing (lines 185-186). For instance, when all legume trees are considered, the significance of the correlation between positive RND and soil N has a p -value of 0.0252 (Supplementary Table 1); however, if only nitrogen-fixing legume trees are considered, the p -value is 0.097 (line 189). Technically, 0.097 is insignificant if the cut off is set to the regular 0.05 level. Does it imply that the nitrogen-fixing ability of the legume trees is not an important factor in this research?

R: As in response to a comment above, in a previous study (Xu, Detto et al., 2019) we showed that neighbors of legumes with high nitrogen fixing abilities showed more positive ^{15}N isotopes signature in the leaf material, therefore the nitrogen-fixing ability of the legume trees is an important factor.

In the discussion, we mention that the decay of the association from p -value 0.0252 to 0.097 is mainly caused by one plot (Rabi) which has a much lower proportion of N-fixers, and where almost 91% of the legume species are non-N-fixers with only four N-fixing species. When data from the Rabi plot are removed, the proportion of legume species with positive RND still shows a significant and strong response to soil total N ($R = 0.854$, $p = 0.0017$, 10 plots). This suggested that the nitrogen-fixing ability of legume species is a critical factor.

(2) Furthermore, the nitrogen-fixing capacity, growth rate, growth character (e.g. shading effects), nutrient requirements (competition effects), microbes at the rhizosphere could be different from one species to another. These factors are ignored in this analysis. How to control for these factors?

R: We agree that all these factors might be important to determine the species composition of around each legume. The scope of our analysis was to determine if legumes have a different effect on neighbors compared to non-legume trees as a first logical step to explore the role of legumes in tropical forests. Once we establish that legumes have different effects compare to non-legume trees, we can ask what causes these differences and how different legumes are expected to influence their neighbors via microbiome or rhizosphere effects. This second step will require to collect legume-specific traits and to intensify the sampling at a single plot scale. We have added these considerations for future directions (L216-233).

(3) Legumes occupy between 0.8% and 74.4% of plot basal area (line 197). Will the RND calculation affected by this factor. For example, when comparing 0.8% occupancy versus 74.4% occupancy, will the estimation of the diversity of non-leguminous trees be affected?

R: The calculation of RND is performed at tree level and averaged across individuals of each species in each plot. Therefore, the estimation of RND should not be biased by the abundance of legume. However, the uncertainty around the estimated value will depend on the species sample size and this might affect the *p*-values of the regression analyses. This problem was accounted for in the randomization test presented in Supplementary Fig. 1.

(4) The comparison of RND among different plots could be problematic. The “non-legume neighborhoods” could be very different in different plots, since the non-legume tree species will be different in different latitudes. Will different non-legume trees in the “non-legume neighborhoods” exert differential effects on the diversity? How to control for this factor?

R: This question is connected to the previous comment. To take into account possible confounding factors among plots (e.g. differences in stem density, basal area etc.) and within plots (e.g. habitat heterogeneity), the diversity around each legume was normalized by the diversity computed for all non-legume trees with similar DBH growing in the same habitat. We refer to this metric as relative neighborhood diversity (RND) to differentiate it from an absolute value of diversity, which would depend on site-specific and local conditions (as explained by the referee). We have improved the description of this in the methods to clarify this point (L285-288).

(5) The soil P and N data are based on the average data from an archive which used a very large grid compared to the current study. Local P and N data in the plots may have changed

due to the species in the legume neighborhoods and non-legume neighborhoods. Local P and N data in the plots should be used.

R: We have answered this question in the first responses to reviewer 1. We repeat that ideally, one should use values of N and P measured in each plot. Unfortunately, these measurements were not available, so we used data from the ORNL DAAC. These data are well maintained and a premier source for global soil nutrient data which are commonly used to perform global analyses.

(6) The authors reported that in 3 plots, “the local soil nutrients in the 0-10 cm layer and those obtained from the global dataset were positively correlated” (lines 243-244). How about the actual data (not just the correlation)? Are they similar and to what degree are the error margins?

R: This is also explained in answer to the first question of reviewer 1. For the local soil nutrients and those from ORNL DAAC, we do not have the soil data from the same depth. For local soil N used, it is sampled at 0-10 cm depth. For ORNL DAAC data, it is from 0-100 cm depth. So, we cannot compare the actual data, and with only three points, we refrain from making conclusions. The maps have been extensively validated using global datasets (<https://daac.ornl.gov>).

(7) The authors concluded that “variation in soil N concentration mediates the relationship between leguminous trees and neighbor diversity but not basal area.” (lines 160-161). “In soils with high N concentration.....but also through N cycling via ectomycorrhizal fungi.....in low soil N levels the presence of N-fixing legumes likely intensifies competition, thereby reducing neighbor diversity.” (lines 168-175).

First, different legume trees will have different associated ectomycorrhizal fungi, the N cycling abilities and the effects on neighbor diversity should be validated if not documented before. Second, why is the competition in low soil N levels only reduce neighbor diversity but not basal area?

R: As mentioned above, we have documented the role of the fixation ability on local diversity in a previous study, using an isotope method (Xu, Detto et al., 2019). Our main finding was that low-N tolerant legumes growing in low soil N habitats display more moderate biological nitrogen fixation (BNF) activity, while high-N demanding legumes growing in high soil N

habitats display higher BNF activity. Our overarching hypothesis is that microbial symbionts can mediate partitioning of soil N and contribute to complementary resource use. Therefore, the presence of fixing legumes in high soil N habitats can facilitate coexistence and increase the local diversity, while in low soil N habitats, the local diversity may decrease because of intensifying competition. Therefore, this hypothesis, named resource demand differentiation, predicts a positive relationship between legume BNF and soil N, and a positive effect of BNF on local diversity. This hypothesis is then verified by the fact that the levels of BNF activity are tightly correlated with legume association to available soil nitrogen, where legumes associated with nitrogen-rich habitats exhibit surprisingly greater BNF activity and a more diverse neighborhood. This hypothesis is also supported by the fact that the BNF activities are often associated with ectomycorrhizal fungi.

For the question about why competition in low soil N habitats only reduces neighbor diversity but not basal area, we think the neighbor biomass (measured as basal area) may be affected by other factors rather than soil N, such as water availability or patch-age. In this study, we tried to find such factors that could be used to explain how neighbor diversity of legumes varied and found one. But for neighbor biomass, we may need to consider a broader spectrum of abiotic and biotic factors (e.g., fungal network connectivity), which involve more-complex, higher-order interactions and symbiotic relationships. Therefore, we have added this as a future direction in the final paragraph of the discussion (L. 228-231).

(8) The authors should add sub-headings to the sections Introduction, Results, and Discussions.

R: added as suggested.

REVIEWERS' COMMENTS:

Reviewer #1 (Remarks to the Author):

In the revised version of the ms, authors have been accepted the limitations of their study, including some quotes into it. From my point of view, I am happy with this version and I have no any further comments on the ms.

Reviewer #3 (Remarks to the Author):

1. The authors suggested that the data in Rabi is the major cause of the decay of p-value 0.0252 to 0.097 when nitrogen-fixing legumes were considered instead of all legumes. First of all, all data, (including the RND for N-fixers, number of N-fixers in each plot, correlation of RND with soil N, etc) mentioned in this paragraph (Line 192-203) should be shown as figures or table in Results instead of brief statements in Discussion. Furthermore, when considering only N-fixing legumes, were the non-fixing legume being treated as "non-legume" or discarded from the calculation? Moreover, the small number of N-fixing species in Rabi is not sufficient to explain this phenomenon when the RND according to the description was already calculated using the N-fixers only. It will be better to supply more explanations for the removal of Rabi from the analysis to obtain a significant correlation.

2. Although a correlation was observed between the soil N and the neighbor diversity of legumes trees, there is no evidence that it is a cause and consequence relationship. The authors need to further tune down the manuscript.

3. Line 259-264: A table should be used to directly show and compare the actual measurement of soil N and P with the global dataset to convince the audience of the correlations.

Responses to Reviewers

Thanks a lot for the constructive comments from editors and referees. All changes are marked with red in the revised manuscript. The comments are also answered one by one as below.

Responses to Reviewer #3

1. The authors suggested that the data in Rabi is the major cause of the decay of p-value 0.0252 to 0.097 when nitrogen-fixing legumes were considered instead of all legumes. First of all, all data, (including the RND for N-fixers, number of N-fixers in each plot, correlation of RND with soil N, etc) mentioned in this paragraph (Line 192-203) should be shown as figures or table in Results instead of brief statements in Discussion. Furthermore, when considering only N-fixing legumes, were the non-fixing legume being treated as “non-legume” or discarded from the calculation? Moreover, the small number of N-fixing species in Rabi is not sufficient to explain this phenomenon when the RND according to the description was already calculated using the N-fixers only. It will be better to supply more explanations for the removal of Rabi from the analysis to obtain a significant correlation.

R: The paragraph about N-fixing legumes analysis is moved from the discussion to the last paragraph of result part and Fig. 5 is added.

In L. 294-296, we illustrated that we calculated the neighbor diversity for all focal species by matching all non-focal species that grow within 60 m from a focal individual and in the same DBH class. The focal species is N-fixing legumes and non-focal species are non-legumes around. To follow the same calculation procedure, when considering only N-fixing legumes, the non-fixing legume is discarded from the calculation.

More explanations for why Rabi plots results are moved from the analysis is added in the discussion part, L. 206-210.

2. Although a correlation was observed between the soil N and the neighbor diversity of legumes trees, there is no evidence that it is a cause and consequence relationship. The authors need to further tune down the manuscript.

R: The causal relationship of the soil N and the neighbor diversity of legumes trees is mainly discussed in the second paragraph in the discussion part. We tone down this paragraph as suggested, L194-197.

3. Line 259-264: A table should be used to directly show and compare the actual measurement of soil N and P with the global dataset to convince the audience of the correlations.

R: Supplementary Table 3 is added.